# Non-Invasive Mapping for Effective Preoperative Guidance to Approach Highly Language-Eloquent Gliomas—A Large Scale Comparative Cohort Study Using a New Classification for Language Eloquence

**DOI:** 10.3390/cancers13020207

**Published:** 2021-01-08

**Authors:** Sebastian Ille, Axel Schroeder, Lucia Albers, Anna Kelm, Doris Droese, Bernhard Meyer, Sandro M. Krieg

**Affiliations:** Department of Neurosurgery & TUM Neuroimaging Center, School of Medicine, Klinikum rechts der Isar, Technical University of Munich, 81675 Munich, Germany; Sebastian.Ille@tum.de (S.I.); Axel.Schroeder@tum.de (A.S.); lucia.albers@tum.de (L.A.); Anna.Kelm@tum.de (A.K.); Doris.Droese@tum.de (D.D.); Bernhard.Meyer@tum.de (B.M.)

**Keywords:** awake surgery, classification, glioma, language, transcranial magnetic stimulation

## Abstract

**Simple Summary:**

Many gliomas are located within highly eloquent areas of language processing, necessitating awake surgery. This study actually proves that the resection of such gliomas can also be performed without awake surgery in two out of three cases, due to preoperative non-invasive mapping by navigated repetitive transcranial magnetic stimulation. Functional and radiological outcome parameters were comparable in both groups. Moreover, we present and validate a newly developed literature-based classification system for language eloquence of brain tumors. Such a classification will enable determining and comparing the language-eloquence of tumor localizations clinically and scientifically, which has not been possible until today due to the heterogeneity of cerebral language and functional reorganization.

**Abstract:**

*Objective:* A considerable number of gliomas require resection via direct electrical stimulation (DES) during awake craniotomy. Likewise, the feasibility of resecting language-eloquent gliomas purely based on navigated repetitive transcranial magnetic stimulation (nrTMS) has been shown. This study analyzes the outcomes after preoperative nrTMS-based and intraoperative DES-based glioma resection in a large cohort. Due to the necessity of making location comparable, a classification for language eloquence for gliomas is introduced. *Methods:* Between March 2015 and May 2019, we prospectively enrolled 100 consecutive cases that were resected based on preoperative nrTMS language mapping (nrTMS group), and 47 cases via intraoperative DES mapping during awake craniotomy (awake group) following a standardized clinical workflow. Outcome measures were determined preoperatively, 5 days after surgery, and 3 months after surgery. To make functional eloquence comparable, we developed a classification based on prior publications and clinical experience. Groups and classification scores were correlated with clinical outcomes. *Results:* The functional outcome did not differ between groups. Gross total resection was achieved in more cases in the nrTMS group (87%, vs. 72% in the awake group, *p* = 0.04). Nonetheless, the awake group showed significantly higher scores for eloquence than the nrTMS group (median 7 points; interquartile range 6–8 vs. 5 points; 3–6.75; *p* < 0.0001). *Conclusion:* Resecting language-eloquent gliomas purely based on nrTMS data is feasible in a high percentage of cases if the described clinical workflow is followed. Moreover, the proposed classification for language eloquence makes language-eloquent tumors comparable, as shown by its correlation with functional and radiological outcomes.

## 1. Introduction

The microsurgical resection of gliomas requires two major aims. On the one hand, the maximization of the extent of resection (EOR) is the determining first step of an optimal oncological treatment [1,2]. On the other hand, the patient’s functionality must always be preserved, and each resection has to avoid permanent surgery-related deficits. To achieve these two paradigms, techniques for the identification of eloquent structures have to be applied with reason. Direct electrical stimulation (DES) during awake craniotomy defines the gold standard technique for cortical and subcortical mapping of functionally eloquent tissue, especially with respect to language function [3,4]. Compared to the results of DES, navigated repetitive transcranial magnetic stimulation (nrTMS) has evolved to be a reliable tool for the non-invasive determination of language-negative sites [5,6,7,8]. The combination of nrTMS-based regions of interest (ROIs) with tractography algorithms has been shown to be an option for the visualization of the subcortical language network, since its correlation with the clinical status of patients has been approved [9,10]. Intraoperative neurophysiology, starting with nrTMS and its successful combination as an adjunct and guide for awake craniotomies, has recently been demonstrated [11]. Thus, nrTMS can help identify patients requiring awake DES mapping and monitoring, while others can be operated on based on the acquired preoperative nrTMS data alone. Smaller cohort studies have shown the feasibility of resections purely based on nrTMS language mapping as a rescue strategy when awake mapping is not available [8,12,13].

The cortical and subcortical language network and its individually associated structures and areas, particularly on the cortical level, is rather complex [14,15]. Due to its complexity, and the additional impact of tumor-induced functional reorganization and plasticity, it is difficult to define single tumors as clearly language eloquent. Although the localization of language function has repeatedly been examined by DES during awake craniotomy in large cohorts, a standardized classification for language eloquence is not available. Even highly standardized and generally accepted scales, such as the Spetzler–Martin grading for brain arteriovenous malformations, avoid a clear rating of language function; nrTMS, meanwhile, was able to prove its capacity to allow an objective definition of eloquence [16].

The present study’s hypothesis is that the resection of language-eloquent gliomas purely based on nrTMS language mapping in a large cohort should show similar functional and radiological outcomes compared to a cohort of patients who underwent DES-based glioma resection during awake craniotomy. 

For the testing of this hypothesis, and to evaluate the presented approach for a function-guided resection of language-eloquent gliomas, we developed a classification of language eloquence in order to make potentially eloquent tumors more comparable. Thus, the second hypothesis is that our newly developed classification of language eloquence should enable us to compare language-eloquent tumors and correlates with the functional and radiological outcomes.

## 2. Methods

### 2.1. Ethics

The study was approved and publicly registered with our university’s ethics board in 2014 (registration number: 222/14). The study was performed in accordance with the Declaration of Helsinki. All included patients provided written informed consent prior to enrolment.

### 2.2. Eligibility Criteria

We prospectively enrolled patients with suspected language-eloquent brain tumors as defined by preoperative magnetic resonance imaging (MRI), and based on the impression of eloquence by the responsible neurosurgical team or the interdisciplinary tumor board. Only patients suffering from anatomically language-eloquent tumors within or adjacent to the classical Broca’s, Wernicke’s, or Geschwind’s area and language-eloquent subcortical pathways who were scheduled for microsurgical resection at our department were enrolled. Indication for resection was made by the interdisciplinary tumor board. In case of the absence of language-positive sites in terms of nrTMS language mapping within the tumor or infiltration zone, the resection was performed purely based on nrTMS data. Otherwise, the indication for an intraoperative DES language mapping was made (Figure 1). Patients with an age of less than 18 years or with severe aphasia, making language mapping impossible, were excluded. Patients with general MRI or TMS exclusion criteria such as cochlear implants or pacemakers were also excluded [17].

### 2.3. Study Protocol

Patients who met the inclusion criteria underwent preoperative nrTMS language mapping prior to surgery. The nrTMS language mapping and tumor resections were performed based on the same structural MRI scan (3T MR scanner Achieva 3T, Philips Medical System, Netherlands B.V.), and according to the standard MRI protocol. For nrTMS-based diffusion tensor imaging fiber tracking (DTI FT), DTI sequences with 32 orthogonal sequences were performed in all patients. The same MRI scan was performed postoperatively within 48 h after surgery for the determination of the EOR. A threshold of <5% of residual tumor was defined to differentiate between gross total resection (GTR) and subtotal resection (STR) [18,19]. The option of an intraoperative MRI (iMRI) was available beginning in March 2018. 

All preoperative nrTMS language mappings were performed according to the standard protocol, using a standard object-naming task (ON) with black-and-white drawings of common objects [20]. After the analysis of elicited naming errors (no response, performance, hesitation, neologism, semantic, phonological, and circumlocution) by the comparison of baseline ON with ON during nrTMS stimulation, we performed nrTMS-based DTI FT using our standard deterministic algorithm [21]. Both cortical language-positive sites based on nrTMS language mapping and nrTMS-based tractography were displayed on the neuronavigation screen during surgery in all cases (Figure 2).

Intraoperative language mappings during awake craniotomy were performed according to our standard asleep-awake-asleep protocol and as recommended by highly experienced groups [22,23]. For the cortical mapping, we used a bipolar stimulation electrode (3–6 mA), and for the subcortical mapping, we used a monopolar stimulation electrode (Inomed Medizintechnik, Emmendingen, Germany). The same ON pictures were used for the preoperative and the intraoperative language mapping procedures. In contrast to the preoperative language mapping, the matrix sentence “This is a…” was used for the intraoperative language mapping.

Clinical language function was assessed preoperatively, five days after surgery, and three months after surgery. For the determination of clinical language function, we rated each patient according to our standard classification system as adapted from the Aachener Aphasie Test (0 = no impairment of language function; 1 = slight impairment of daily communication; 2 = moderate impairment of language function, daily communication possible; 3 = severe impairment of language function, daily communication not possible; A = non-fluent; B = fluent) [6].

### 2.4. Classification of Language Eloquence

To determine the language eloquence of gliomas and to compare patients who underwent nrTMS-based resection with patients who underwent DES-based resection, we performed a literature and database search for a classification of language eloquence on PubMed, MEDLINE, and the Cochrane Library. None of the publications described a comprehensive classification for language eloquence. 

Afterward, the same databases were searched for relevant publications of DES-based language mappings [4,14,15,24,25,26]. Based on these publications we developed a new three-tier classification that sums cortical (Co), subcortical (S), and clinical (Cl) characteristics of language eloquence. Each of the characteristics is subdivided into high, moderate, and low probabilities of language eloquence, leading to zero, one (^1^), two (^2^), or three (^3^) points per characteristic. After the addition of the three subdivisions, the final grading reveals a low (<3 points), moderate (3–5 points), or high (>5 points) grade of language eloquence. Table 1 summarizes the classification. The eloquence of the tumor localization is determined based on the preoperative MRI scan without DTI FT. The relation to white matter pathways is rated according to the anatomical determination of the surgeon. Cortical eloquence is defined by the localization of the tumor within the according area, or as described by a distance in the table. Subcortical eloquence is defined by a distance of <5 mm between the tumor and the according white matter pathway, or as described by a distance in the table.

The table shows the subdivision for the initial rating and the according points of the new language classification. After the addition of the three subdivisions the final grading reveals a low (<3 points), moderate (3–5 points), or high (>5 points) grade of language eloquence. The eloquence of the tumor localization is determined based on the preoperative MRI scan without diffusion tensor imaging fiber tracking (DTI FT). The relation to the white matter pathways is rated according to the anatomical determination of the surgeon. Cortical eloquence is defined by the localization of the tumor within the according area, or as described by a distance in the table. Subcortical eloquence is defined by a distance of <5 mm between the tumor and the according white matter pathway, or as described by a distance in the table.

### 2.5. Statistical Analysis

All of the analyses were performed using the GraphPad Prism software (GraphPad Prism 8, San Diego, CA, USA). A *p*-value of less than 0.05 was considered statistically significant. Initially, a Gaussian distribution was tested for all measures. The two groups’ baseline characteristics were compared using independent t-tests for continuous variables, and with Fisher’s exact or chi-square tests for the categorical variables. If the null hypothesis was rejected based on a *p*-value < 0.05, further calculations for the tested data were performed using the Mann–Whitney test. In case of no rejection of the null hypothesis based on a *p*-value > 0.05, further calculations for the tested data were performed using both parametric and non-parametric tests. In these cases, the manuscript and tables show the *p*-value results of the *t*-tests.

## 3. Results

### 3.1. Patient Characteristics

Between March 2015 and May 2019, we included 147 consecutive cases (68 female, 79 male) with a mean (±standard deviation) age of 54 ± 15 (minimum–maximum 20–84) years. The histopathological diagnosis of a glioma was confirmed in all cases. The tumors were recurrent gliomas in 60 cases (40.8%). The gliomas were located within the left hemisphere in 143 cases (97.3%). Language mappings of the patients with right-hemispheric gliomas (4 cases, 2.7%) were exclusively performed by nrTMS. Preoperative clinical symptoms and clear left-handedness, as measured by the Edinburgh Handedness Inventory, were used to reason the preoperative language mapping of patients with right-hemispheric tumors. In total, we performed language mappings of more than one language in 11 bilingual patients (7.5%).

To summarize all 147 cases, the patients did not show new language deficits postoperatively in 99 cases (67.3%). In 32 cases (21.8%), the patients suffered from transient new language deficits, and in six cases (4.1%), they had permanent, new language deficits after surgery. The language outcome could not be feasibly measured due to the general postoperative status of persistently decreased patient vigilance in 10 cases (6.8%).

### 3.2. Functional and Radiological Outcome

Overall, GTR was achieved in 121 cases (82.3%), and STR was achieved in 26 cases (17.7%). Since an iMRI became available at the department in March 2018, iMRI has been performed in 42 of 77 potential cases (55.8%). In order to subdivide patients who were graded with highly eloquent versus moderately eloquent tumors, we separated the outcome measures and ratings of these two groups of patients. Based on this separation, both groups showed comparable clinical and radiological outcomes (Table 2).

We performed glioma resection purely based on nrTMS language mapping in 100 cases (68.0%). Within the same period, we performed DES-based glioma resection during awake craniotomy in 47 cases (32.0%). Apart from a lower mean age and more insular tumors among the patients in the awake group, the two groups did not show statistically significant differences (Appendix A), including in clinical outcome. We did not find differences between patients suffering from low-grade or high-grade gliomas regarding the language mapping per se, or in clinical and radiological outcome. In contrast, a GTR was achieved more often in the nrTMS group compared to the EOR in the awake group. Appendix A and Figure 3 and Figure 4 show differences in the clinical and radiological outcome measures. Figure 5 shows illustrative cases of pre- and postoperative imaging of patients with low- and high-grade gliomas.

### 3.3. Classification of Language Eloquence

Language eloquence could be determined using the new classification in all cases. Overall, we found a median (interquartile range) language eloquence of 6 (4–7) points. As finally graded by the classification, the overall cohort’s tumors were highly language-eloquent in 76 cases (51.7%), and moderately language-eloquent in 60 cases (40.8%). In 11 cases (7.5%), the localizations of tumors showed low language eloquence. We found statistically significant differences in the rating, sum of points, and final grading between the two groups (Table 3 and Figure 6 and Figure 7). 

## 4. Discussion

### 4.1. Feasibility of nrTMS-Based Glioma Resection

The analysis of the present cohort confirms that the resection of language-eloquent gliomas purely based on nrTMS language mapping is feasible and safe, and supported by similar functional and radiological outcomes compared to those of a cohort of patients who underwent DES-based glioma resection during awake craniotomy. Both the clinical and radiological outcomes of patients in the nrTMS group emphasize this hypothesis with respect to the current literature on DES-based glioma resections, and as compared to our DES cohort’s outcomes [3,4]. Most importantly, the present study is not meant to replace DES-based glioma resections by nrTMS-based resections. However, as the presented results show, the presence of a tumor within or adjacent to language-eloquent regions does not disqualify it from a resection purely based on nrTMS. Despite the largest proportion of patients, who underwent DES-based resection during awake craniotomy, being graded as having highly eloquent gliomas (85.1%), we also found a notable proportion of high gradings in the nrTMS group (36.0%). Neither the EOR nor the clinical outcome significantly differed from the results of highly graded patients who underwent DES-based resection during awake craniotomy (Table 2). As expected, we found a statistically significant difference in EOR between the two groups. A GTR was achieved in more cases in the nrTMS group than in the awake group. On the one hand, this is obvious since patients who underwent DES-based glioma resection suffered from tumors with higher eloquence (Table 2 and Table 3, Figure 6 and Figure 7). On the other hand, by using the classification, we were able to show that the main part of the included patients (92.5%) suffered from moderately or highly language-eloquent gliomas (Table 3). Basically, we know that the EOR can be extended to the border, which is relevant for preserving functionality, by applying DES. This is supported by the expected higher rate of transient deficits in the awake group (31.9% vs. 17.0%; Appendix A). Nevertheless, the analysis of EOR in combination with the language eloquence of tumors shows that glioma resection purely based on nrTMS language mapping also enables language-eloquent gliomas to be resected based on the current neurosurgical standard for balancing oncology and functionality (Table 2).

We do not recommend that every glioma should be resected purely based on preoperative nrTMS mappings. Intraoperative neuromonitoring and neurophysiology start with preoperative mapping, whose results can additionally be used for a function-based DTI FT approach for visualizing subcortical white matter pathways. Furthermore, the results of preoperative mappings can support the intraoperative mapping procedure [11]. Obviously, we still performed a high percentage of glioma resections via awake craniotomies.

### 4.2. Differentiation through Classification of Language Eloquence

The present analysis shows that the newly developed classification of language eloquence allows language-eloquent tumors to be compared, and that the classification is correlated with the functional and radiological outcomes. The classification features attention to vulnerable cortical and subcortical structures, as well as the related appearance of clinical symptoms (Table 1). In particular, the necessity of preserving subcortical language-eloquent white-matter pathways, as shown by resection probability maps, plays a central role, and is reflected in the comparison of the two groups [24,25]. Nearly 90% of patients in the awake group showed a S^3^ rating, meaning a high probability of subcortical language eloquence (Table 1). Additionally, the two groups differed in cortical eloquence, as measured by the higher percentage of Co^3^ ratings, and by the proportion of patients who showed clinical symptoms due to tumor localization or prior resections, as demonstrated by the larger proportion of Cl^2^ ratings (Table 3). The higher percentage of Co^3^ ratings in the awake group is explained by the presented workflow (Figure 1). Similarly, this fact justifies the presented approach and incidentally confirms the accordance of the two techniques. Furthermore, the classification’s reliability is confirmed by the fact that most patients in the awake group (85.1%) were rated at 6 points or more after summing up the single ratings, as reasoned by at least two major ratings (Co^3^ and S^3^) or a rating of eloquence in each of the three subdivisions (Table 1 and Appendix A). In contrast, the largest portion of the nrTMS group (53.0%) was graded with moderate eloquence. Apart from the rating of high grades of eloquence, none of the comparisons showed statistically significant differences (Table 2). Hence, the gradings of high and moderate eloquence showed particular reliability for both the awake and the nrTMS groups.

### 4.3. Limitations

The new classification for language eloquence was used for the first time. Despite it being based on prior publications, the present analysis is its first validation. In particular, the correlation between a higher proportion of permanent deficits and fewer GTRs of high-graded patients confirms the classification’s reliability, validity, and applicability. Furthermore, the comparison of similarly graded patients showed equal postoperative functional outcomes in both groups (Table 2). Nevertheless, we encourage further centers to evaluate the present classification of language eloquence in order to modify it, or to confirm its reliability and applicability.

## 5. Conclusions

The analysis of the present cohort confirms that the resection of language-eloquent gliomas, purely based on nrTMS language mapping, is feasible if the described clinical workflow is followed; the clinical and oncological outcomes are highly comparable to those of awake cohorts. Additionally, classifying language eloquence enables language-eloquent tumors to be compared.

## Figures and Tables

**Figure 1 cancers-13-00207-f001:**
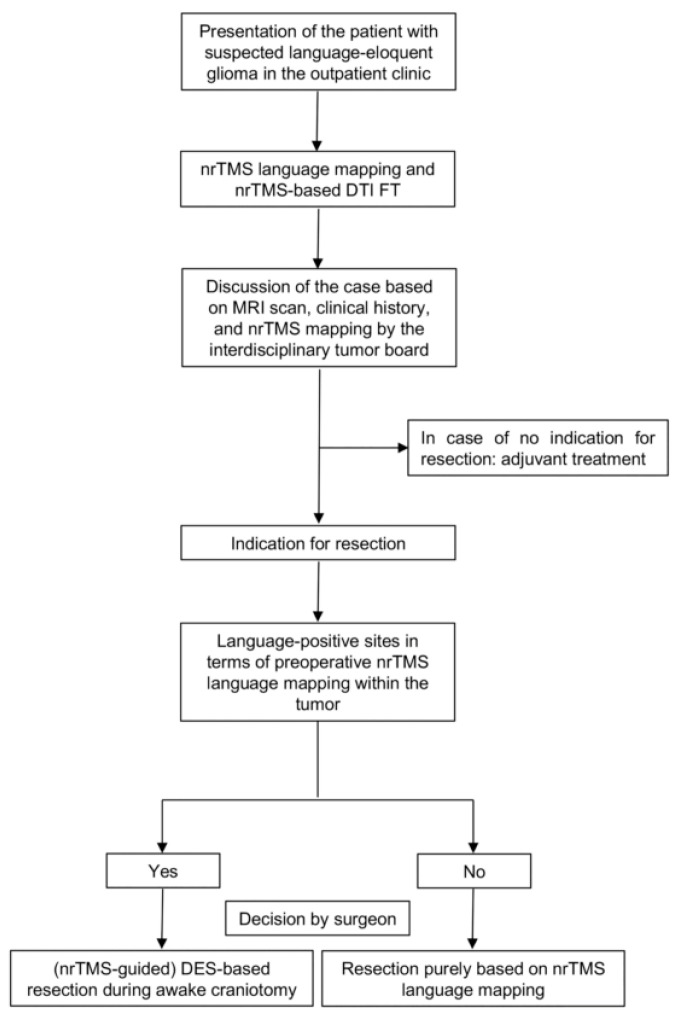
Standard procedure. The flowchart describes the standard procedure for the decision for a DES-based resection during awake craniotomy or for a nrTMS-based resection of a language-eloquent tumor (DTI = diffusion tensor imaging, DTI FT = DTI fiber tracking, nrTMS = navigated repetitive transcranial magnetic stimulation, DES = direct electrical stimulation).

**Figure 2 cancers-13-00207-f002:**
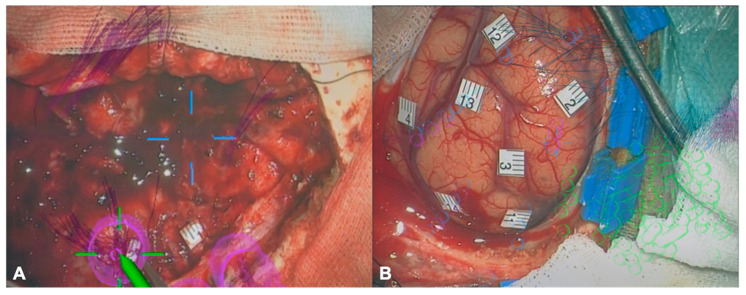
Overlay of language-positive sites. The figure shows illustrative cases of the accordance of language-positive sites as mapped by nrTMS language mapping (pink sites) and DES language mapping during awake craniotomies (white single digit platelets) shown by the visual overlay of the microscope view after resection of a left fronto-insular glioblastoma (**A**), and before resection of a left parietal glioblastoma (**B**). Blue (nrTMS mapping of arithmetic processing) and green (nTMS mapping of motor function) sites as well as white double-digit platelets (**B**) have not been considered for the present analysis.

**Figure 3 cancers-13-00207-f003:**
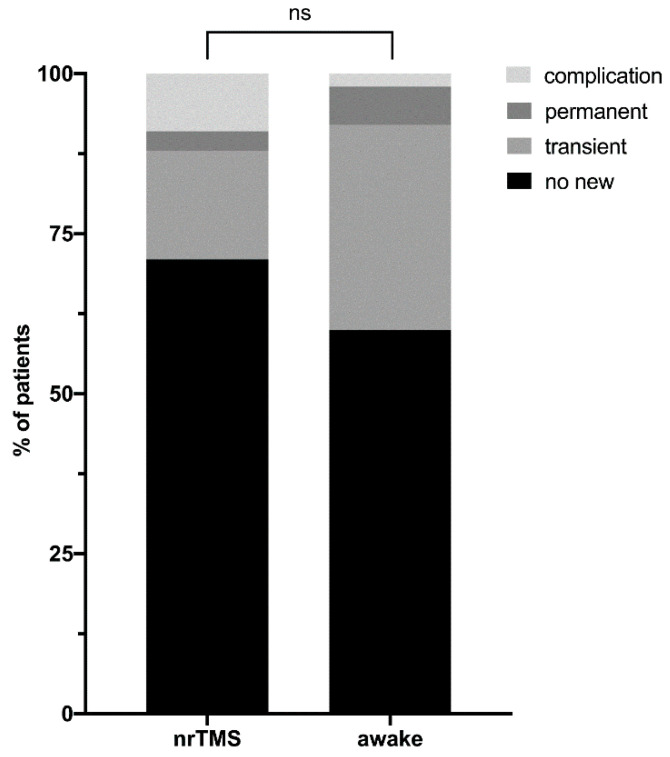
Language outcome. The figure summarizes the comparison of language outcomes of the two groups. Transient language deficits were defined as new surgery-related aphasia, as examined five days after surgery, but the new aphasia was not persistent three months after surgery. Permanent language deficits were defined as new surgery-related aphasia, as examined five days after surgery and three months after surgery. Complication describes cases in both groups in which the examination of the language outcome was not feasible due to a persistently decreased vigilance.

**Figure 4 cancers-13-00207-f004:**
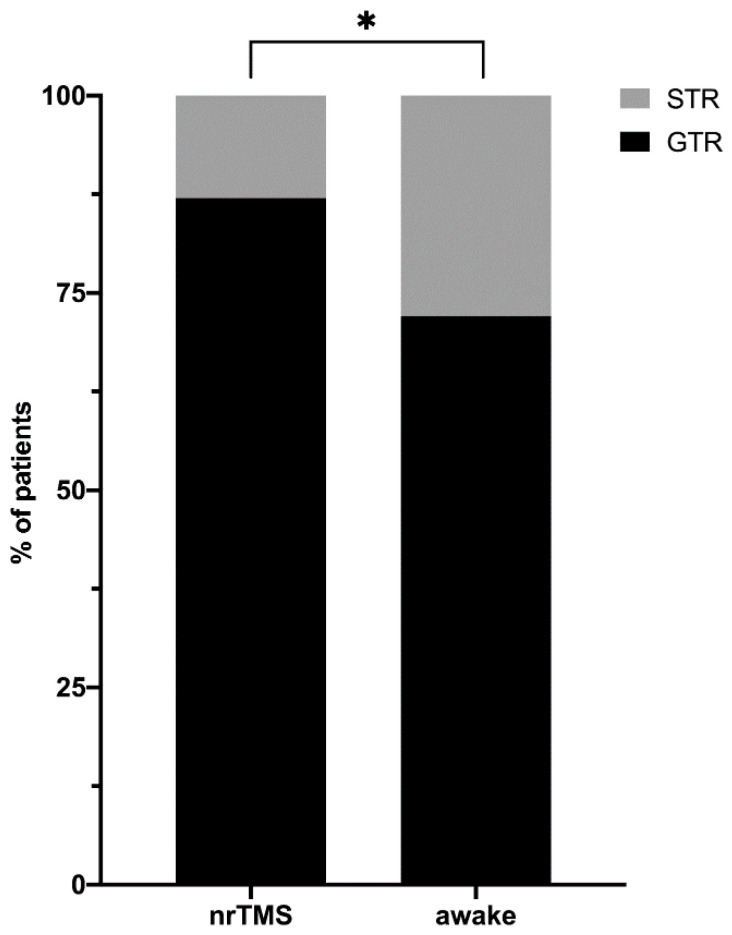
Extent of resection. The figure summarizes the comparison of radiological outcomes of the two groups. The threshold for the differentiation between GTR and STR was 95% of the initial tumor volume (GTR = gross total resection, STR = subtotal resection, * = *p* < 0.05).

**Figure 5 cancers-13-00207-f005:**
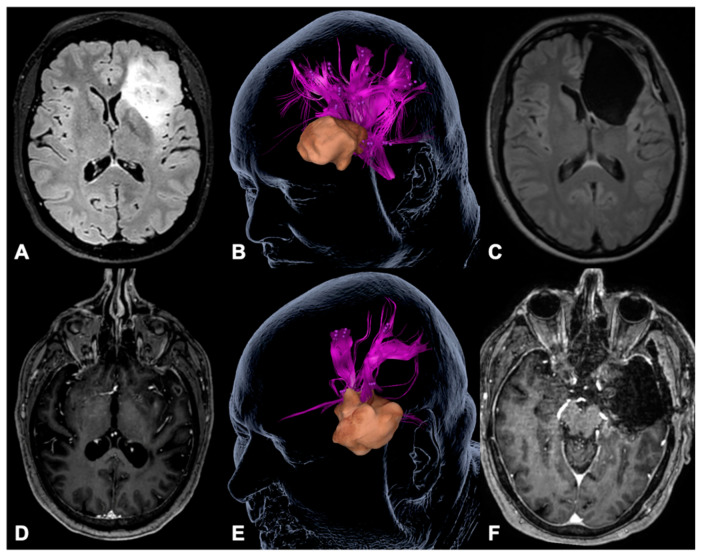
Illustrative cases. The figure shows an illustrative case (**A**–**C**) of a patient suffering from a left fronto-insular diffuse astrocytoma WHO II (**A**), who underwent gross total tumor resection (**C**) based on DES language mapping during awake craniotomy after we found language-positive cortical sites in terms of nrTMS language mapping within the opercular part of the inferior frontal gyrus (pink sites, **B**). Clinically, the patient suffered from a slight transient fluent aphasia, grade 1B, postoperatively. The second patient (**D**–**F**) suffered from a left temporo-insular glioblastoma (**D**) and underwent gross total tumor resection (**F**) purely based on nrTMS language mapping data (**E**). The patient suffered from a slight transient non-fluent aphasia, grade 1A, postoperatively. Both patients did not show any permanent functional deficits.

**Figure 6 cancers-13-00207-f006:**
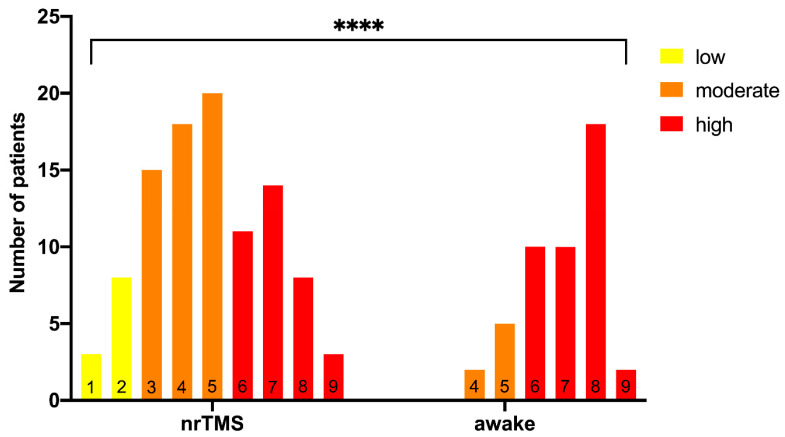
Points and grading. The figure shows the summarized points after the rating of the two groups and the final grading of all patients (low = less than 3 points, moderate = 3–5 points, high = more than 5 points, **** = *p* < 0.0001).

**Figure 7 cancers-13-00207-f007:**
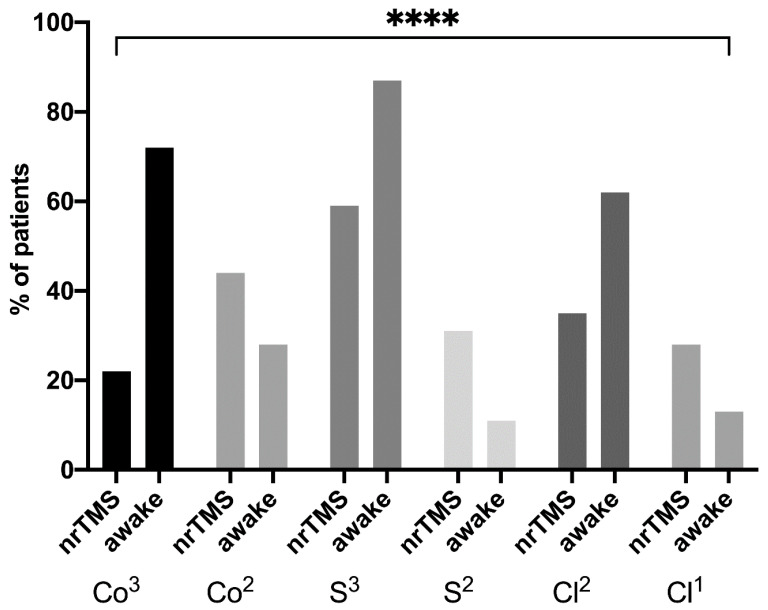
Rating. The figure summarizes the distribution of the ratings of language eloquence of the two groups (Co^3^ = high cortical, Co^2^ = moderate cortical, S^3^ = high subcortical, S^2^ = moderate subcortical, Cl^2^ = high clinical, Cl^1^ = moderate clinical, **** = *p* < 0.0001).

**Table 1 cancers-13-00207-t001:** Classification of language eloquence.

Classification			Points
High	Cortical	Opercular inferior frontal gyrus	3
		Posterior supramarginal gyrus	
		Angular gyrus	
		Posterior middle frontal gyrus	
		Posterior superior temporal gyrus	
		Middle superior temporal gyrus	
	Subcortical	Arcuate fasciculus/deep superior longitudinal fasciculus	
		Superior longitudinal fasciculus II & III	
		Inferior fronto-occipital fasciculus	
		Uncinate fasciculus	
	Clinical	Preoperative language deficit * due to tumor growth	2
		Postoperative language deficit * after prior resection **	
Moderate	Cortical	Triangular inferior frontal gyrus	2
		Anterior supramarginal gyrus	
		Middle middle frontal gyrus	
		Posterior middle temporal gyrus	
		Middle middle temporal gyrus	
		Posterior superior frontal gyrus	
		<5 mm to Co^3^	
	Subcortical	Middle longitudinal fasciculus	
		Inferior longitudinal fasciculus	
		5–10 mm to S^3^	
	Clinical	Focal seizure accompanied by language deficit	1
Low	Cortical	>5 mm to Co^3^	0
		Not within Co^2^	
	Subcortical	>10 mm to S^3^	
		>5 mm to S^2^	
	Clinical	No clinical history of language impairment	

* transient or permanent; ** non-vascular, non-complication; Co^3^ = high cortical, Co^2^ = moderate cortical, S^3^ = high subcortical, S^2^ = moderate subcortical, Cl^2^ = high clinical, Cl^1^ = moderate clinical.

**Table 2 cancers-13-00207-t002:** Comparison of high and moderate gradings.

Grading	Cases		nrTMS	Awake	*p*-Value
			36 (36.0)	40 (85.1)	
Grading high	Rating	Co^3^	18 (50.0)	34 (85.0)	0.0228
		Co^2^	17 (47.2)	6 (15.0)	
		S^3^	32 (88.9)	36 (90.0)	
		S^2^	4 (11.1)	4 (10.0)	
		Cl^2^	25 (69.4)	28 (70.0)	
		Cl^1^	13 (36.1)	6 (15.0)	
	Sum of points	6	11 (30.6)	10 (25.0)	0.2022
		7	14 (38.9)	10 (25.0)	
		8	8 (22.2)	18 (45.0)	
		9	3 (8.3)	2 (5.0)	
	Outcome	no new	25 (69.4)	22 (55.0)	0.4205
		transient	7 (19.4)	14 (35.0)	
		permanent	2 (5.6)	3 (7.5)	
		complication	2 (5.6)	1 (2.5)	
	EOR	GTR	30 (83.3)	28 (70.0)	0.1903
		STR	6 (16.7)	12 (30.0)	
	Cases		53 (53.0)	7 (14.9)	
Grading moderate	Rating	Co^3^	4 (7.5)	0	0.2474
		Co^2^	25 (47.2)	7 (100)	
		S^3^	27 (50.9)	5 (71.4)	
		S^2^	21 (39.6)	1 (14.3)	
		Cl^2^	10 (18.9)	1 (14.3)	
		Cl^1^	12 (22.6)	0	
	Sum of points	3	15 (28.3)	0	0.1725
		4	18 (34.0)	2 (28.6)	
		5	20 (37.7)	5 (71.4)	
	Outcome	no new	36 (67.9)	6 (85.7)	0.9253
		transient	10 (18.9)	1 (14.3)	
		permanent	1 (1.9)	0	
		complication	6 (11.3)	0	
	EOR	GTR	46 (86.8)	6 (85.7)	>0.9999
		STR	7 (13.2)	1 (14.3)	

The table shows the comparison of ratings and outcomes of patients who were graded with highly or moderately language eloquent tumors. The threshold for the differentiation between GTR and STR was >95% of the initial tumor volume (EOR = extent of resection, GTR = gross total resection, STR = subtotal resection, Co^3^ = high cortical, Co^2^ = moderate cortical, S^3^ = high subcortical, S^2^ = moderate subcortical, Cl^2^ = high clinical, Cl^1^ = moderate clinical).

**Table 3 cancers-13-00207-t003:** Comparison of language classification.

Classification		nrTMS	Awake	*p*-Value
Rating	Co^3^	22 (22.0)	34 (72.3)	<0.0001
	Co^2^	44 (44.0)	13 (27.7)	
	S^3^	59 (59.0)	41 (87.2)	
	S^2^	31 (31.0)	5 (10.6)	
	Cl^2^	35 (35.0)	29 (61.7)	
	Cl^1^	28 (28.0)	6 (12.8)	
Sum of points	1	3 (3.0)		<0.0001
	2	8 (8.0)		
	3	15 (15.0)		
	4	18 (18.0)	2 (4.3)	
	5	20 (20.0)	5 (10.6)	
	6	11 (11.0)	10 (21.3)	
	7	14 (14.0)	10 (21.3)	
	8	8 (8.0)	18 (38.3)	
	9	3 (3.0)	2 (4.3)	
Grading	high	36 (36.0)	40 (85.1)	<0.0001
	moderate	53 (53.0)	7 (14.9)	
	low	11 (11.0)	0	

The table shows the comparison of the results of language classification for the two groups (Co^3^ = high cortical, Co^2^ = moderate cortical, S^3^ = high subcortical, S^2^ = moderate subcortical, Cl^2^ = high clinical, Cl^1^ = moderate clinical).

## Data Availability

The data presented in this study are available on request from the corresponding author. The data are not publicly available due to privacy restrictions. The corresponding author had full access to all the data in the study and had final responsibility for the decision to submit for publication.

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
