# Peer review of "Non-Invasive Mapping for Effective Preoperative Guidance to Approach Highly Language-Eloquent Gliomas—A Large Scale Comparative Cohort Study Using a New Classification for Language Eloquence"

_cancers, 2021, doi:10.3390/cancers13020207_

Round 1

Reviewer 1 Report

The paper describes preoperative non-invasive mapping i.e. „navigated repetitive transcranial magnetic stimulation (nrTMS)“ as a useful tool for resectable, however highly language-eloquent gliomas. The authors have longstanding expertise in the field. In their clinical workflow, they describe their standard procedure when to do awake craniotomy with direct electrical stimulation (DES) and when to resect a tumor purely based on nrTMS findings. In case of the absence of language-positive sites within the tumor or infiltration zone detected by nrTMS, the resection could be performed purely based on nrTMS data. Otherwise, they propose tumors within language mapping to be resected by DES. Postoperative results of 147 consecutive patients endorse this clinical approach.

Comments:

  1. does the title of the paper reflect the author`s approach for highly language –eloquent gliomas? Or should the title rather be: „non-invasive mapping as effective preoperative guidance on how to approach highly language –eloquent glioma“? In the discussion part, the authors state clearly: „results show that nrTMS-based resection is not meant to replace DES-based glioma resections“.
  2. Please comment more clearly on the two cohorts („nrTMS only“ vs „nr TMS“ followed by wake „DES“). They seem to be different in one essential aspect: tumors within language mapping vs not within language mapping zone. Is it possible to compare postoperative results and draw conclusions from these two quite different cohorts?
  3. Point 2 also refers to Figure 3: are these two cohorts really comparable? It seems obvious and predictable that GTR is less likely in the group of tumors within the eloquent language zone?
  4. The second hypothesis and the strength of the paper is a proposal for a new classification for language eloquence. The authors have to be congratulated for this initiative. As they correctly state, this classification has to be validated by other research groups.
  5. Contradictions in the discussion part should be explained in more detail: e.g. page 10, line 237 „results show that nrTMS-based resection is not meant to replace DES-based glioma resection. Nevertheless, the presence of a tumor within or adjacent to language-eloquent regions does not disqualify it from a resection purely based on nrTMS“. Where does this conclusion come from, and what means "nevertheless"?

Minor comments:

  1. please insert a legend below Table 2 and 3 as you do in Figure 5
  2. The discussion part could be shortened and limited to the essential points.

Author Response

Reviewer #1

The paper describes preoperative non-invasive mapping i.e. „navigated repetitive transcranial magnetic stimulation (nrTMS)“ as a useful tool for resectable, however highly language-eloquent gliomas. The authors have longstanding expertise in the field. In their clinical workflow, they describe their standard procedure when to do awake craniotomy with direct electrical stimulation (DES) and when to resect a tumor purely based on nrTMS findings. In case of the absence of language-positive sites within the tumor or infiltration zone detected by nrTMS, the resection could be performed purely based on nrTMS data. Otherwise, they propose tumors within language mapping to be resected by DES. Postoperative results of 147 consecutive patients endorse this clinical approach.

Comment 1:

Does the title of the paper reflect the author`s approach for highly language –eloquent gliomas? Or should the title rather be: „non-invasive mapping as effective preoperative guidance on how to approach highly language –eloquent glioma“? In the discussion part, the authors state clearly: „results show that nrTMS-based resection is not meant to replace DES-based glioma resections“.

Answer 1:

We thank you for your comment. We rephrased the title to answer your comment.

Comment 2:

Please comment more clearly on the two cohorts („nrTMS only“ vs „nr TMS“ followed by wake „DES“). They seem to be different in one essential aspect: tumors within language mapping vs not within language mapping zone. Is it possible to compare postoperative results and draw conclusions from these two quite different cohorts?

Answer 2:

Dear reviewer, to answer your comment, we have to clarify two essentials: firstly, all included patients who suffered from anatomically language-eloquent gliomas located within or adjacent to classical Broca’s, Wernicke’s, or Geschwind’s area or language-eloquent subcortical pathways. Secondly, according to our standard protocol and as stated in the Methods section of the manuscript, we performed preoperative nrTMS language mappings of the whole hemisphere and not only of the peritumoral zone. Hence, the two cohorts do not differ between tumors within and not within the language mapping zone. As Figure 1 shows, the decision to perform an awake craniotomy including intraoperative mapping was then made depending on the preoperative nrTMS language mapping results. Furthermore, as shown by the results of the language classification and particularly Table 3, only 11 patients (11% of the nrTMS group, 7.5% of the whole cohort) were graded with low language eloquence. Similarly, 92.5% of all included patients were graded with moderate or high language eloquence. Therefore, even without consideration of the language classification, we only included patients with language-eloquent gliomas and compared their outcomes depending on the differentiation between nrTMS-based or intraoperative awake mapping-based resection. We additionally highlighted this in the methods section of the manuscript to clarify the inclusion criteria.

Comment 3:

Point 2 also refers to Figure 3: are these two cohorts really comparable? It seems obvious and predictable that GTR is less likely in the group of tumors within the eloquent language zone?

Answer 3:

As already described by the answer on comment 2, we only included patients with language-eloquent gliomas. However, these patients underwent resection of language-eloquent gliomas based on two different approaches for language mapping and the detection of functionality (nrTMS-based vs. DES-based during awake craniotomy). Hence, we compared two different approaches in a similar and thereby comparable group of patients regarding their clinical and radiological outcome.

We added the classification of language eloquence to differ between moderate and high language eloquence. As stated in the discussion section of the manuscript, as expected, patients included in the awake group – suffering from gliomas with location of higher eloquence – showed more often a subtotal resection as compared to patients in the nrTMS group. Thus, the justification for a classification of language eloquence came up to make such cohorts comparable.

With the aim of comparing patients with similar tumor locations and similar language eloquence, we additionally analyzed patients of both groups with a high grading and a moderate grading of language eloquence (Table 2).

Comment 4:

The second hypothesis and the strength of the paper is a proposal for a new classification for language eloquence. The authors have to be congratulated for this initiative. As they correctly state, this classification has to be validated by other research groups.

Answer 4:

We thank you for this comment. Of course, the classification has to be validated by other groups. Meanwhile, this process has already been initialized.

Comment 5:

Contradictions in the discussion part should be explained in more detail: e.g. page 10, line 237 „results show that nrTMS-based resection is not meant to replace DES-based glioma resection. Nevertheless, the presence of a tumor within or adjacent to language-eloquent regions does not disqualify it from a resection purely based on nrTMS“. Where does this conclusion come from, and what means "nevertheless"?

Answer 5:

By this sentence we just wanted to state that our group does not aim to replace glioma resection based on DES during awake surgery by nrTMS-based resections. “Nevertheless” means that still, the results of the present study show that language-eloquent gliomas in certain localizations can also be resected purely based on nrTMS language mapping with optimal clinical and radiological outcome. However, in order to answer your comment we rephrased this part (p. 12, l. 290-291).

Minor comments:

Comment 6:

please insert a legend below Table 2 and 3 as you do in Figure 5

Answer 6:

Thank you for this comment. We added the legends explaining the classification abbreviations.

Comment 7:

The discussion part could be shortened and limited to the essential points.

Answer 7:

In order to answer your comment, we shortened the discussion section.

Reviewer 2 Report

Awake craniotomy still remain a gold standard in cases where eloquent  areas are involved. The major  limit of   this methodology is that we can use it  in a quite  limited number of cases,  mainly in the  low grade tumors    even  if we would like to have the same intraoperative feed back in a much larger  number of neurosurgical cases  .  In effect, the preoperative search   of cerebral  functions and its precise location    is  a very important issue . Undoubtedly,  good result has been obtained  for motor-sensitive  central area.

The authors present the use   of  nrTMS  in language  eloquent areas .

Some questions:

  1. The authors simply speak about language without any distinction . The should better define what kind of function they have looked for and which has been the percentage of positive response   for any singular function they have tested.
  2. A very important issue has always been the coincidence between preoperative functional data and intraoperative mapping. It would be very important to know the accuracy of the concidence between the functional location found by the nvTMS , the fMRI and the intraoperative response obtained with mapping of the same test.
  3. The authors should show some images with good and also bad correlation between nvTMS location     and f MRI of the same test using Flair sequences.
  4. They should also clear if there is any difference of response between low grade and high grade lesions
  5. About the resection: they should stratify the volumes of low grade and high grade lesions. It will be useful to see some images of total resection of a low grade and of high grade tumor in regard to the positive response of TMS.
  6. I thing that the classification of eloquence is out of place in this paper and the chapter should be cancelled . The variability of language functions is very high and we need much larger numbers of cases. It is not the case of this paper where the number are low with different histology. The data can’t be reliable enough.

The main issue of this paper is nvTMS and its reliability in surgery. They should highlight this aspect with more detailed data  showing the percentage of positive response of the different function of the language area, their concidence with fMRI and intraoperative mapping  that give to us the level of the guarantee we can offer to the patient.   The authors  have presented their  work in a too descriptive way  which is not acceptable for a publication.  A major  revision with more details   is needed  to reconsider this paper.

Author Response

Reviewer #2

Awake craniotomy still remain a gold standard in cases where eloquent areas are involved. The major limit of   this methodology is that we can use it in a quite  limited number of cases,  mainly in the  low grade tumors even if we would like to have the same intraoperative feedback in a much larger  number of neurosurgical cases.  In effect, the preoperative search   of cerebral functions and its precise location    is  a very important issue. Undoubtedly,  good result has been obtained  for motor-sensitive  central area.

The authors present the use of  nrTMS  in language  eloquent areas .

Some questions:

Comment 1:

The authors simply speak about language without any distinction. They should better define what kind of function they have looked for and which has been the percentage of positive response   for any singular function they have tested.

Answer 1:

Dear reviewer#2, as stated in the methods section of the manuscript, we analyzed clearly defined error categories (no response, performance, hesitation, neologism, semantic, phonological, and circumlocution) (p. 3, l. 112-113). These have been used pre- and intraoperatively. This all aspects of language where covered considering the complexity of language processing of the brain.

This comprehensive kind of analyzing language mappings has been developed by highly experienced groups performing language mappings and their analysis for decades [1]. Furthermore, pre- and intraoperative language mappings as well as their analysis have been performed according to standardized protocols and current standard as described in the methods section [2-4].

Comment 2:

A very important issue has always been the coincidence between preoperative functional data and intraoperative mapping. It would be very important to know the accuracy of the coincidence between the functional location found by the nvTMS , the fMRI and the intraoperative response obtained with mapping of the same test.

Answer 2:

We thank you for this comment. First of all, we did not perform localization of language function by functional MRI (fMRI) in the present study.

As published by our group in 2015 but also many others over the years, the localization of language function by fMRI did not show accordance with the gold standard technique DES language mapping during awake craniotomy in a large group of patients with language-eloquent brain lesions [5]. Pathological vascularization within and adjacent to the tumor might be the reason for the impairment of blood oxygen level dependent (BOLD) signals in brain tumor patients as already published by other groups [6].

As explained in the Introduction section of the manuscript, several groups including ours have meanwhile published the reliability of nrTMS language mapping for the localization of language-negative brain regions. This reliability is based on large cohort comparisons to the gold standard technique DES language mapping during awake craniotomy [7-10]. The presented approach as shown in Figure 1 is based on the high negative predictive value but low positive predictive value of nrTMS language mapping as compared to DES language mapping during awake craniotomy. In case of language-positive sites in terms of nrTMS language mapping within or adjacent to the tumor, we performed an awake craniotomy to review these language-positive sites intraoperatively using DES. In order to answer your comment, we added Figure 2.

Comment 3:

The authors should show some images with good and also bad correlation between nvTMS location and f MRI of the same test using Flair sequences.

Answer 3:

Dear reviewer, as already explained in Answer 2, we did not use localization of language function by fMRI in the present study.

We never mentioned using fMRI.

We performed language mapping by DES during awake craniotomy and nrTMS as described in the methods section of the manuscript.

Comment 4:

They should also clear if there is any difference of response between low grade and high grade lesions

Answer 4:

If we understand you correctly, your question for response is meant in terms of receiving reliable language mapping results in patients suffering from low-grade or high-grade gliomas. Numbers of included patients suffering from low-grade or high-grade gliomas are shown in Supplementary Table 1. Apart from that we did not find differences regarding the outcome of patients with low-grade or high-grade gliomas. To answer your comment, we added this statement to the manuscript.

Comment 5:

About the resection: they should stratify the volumes of low grade and high grade lesions. It will be useful to see some images of total resection of a low grade and of high grade tumor in regard to the positive response of TMS.

Answer 5:

In order to answer your comment we added Figure 5.

Comment 6:

I think that the classification of eloquence is out of place in this paper and the chapter should be cancelled . The variability of language functions is very high and we need much larger numbers of cases. It is not the case of this paper where the number are low with different histology. The data can’t be reliable enough.

Answer 6:

Dear reviewer, as you know and as it has been published by leading groups performing language mapping and research for decades, the variability particularly of the cortical localization of human language function is tremendously and highly individual in the single patient due to glioma- and surgery-induced functional reorganization, effects of plasticity, and further impacts [11,12]. Hence, the localization of cortical language function cannot be reduced on classical anatomical sites but depends on our current knowledge which bases on intraoperative cortical language mappings and the analysis of language-associated white matter pathways as performed by leading groups [11-15]. Even more, the preoperative impact of tumor growth and the history of prior resections impacts the eloquence of tumor localization.

Therefore, the classification of language eloquence has been applied in order to analyze this large cohort of patients comprehensively. Without using any description of language eloquence we could not differentiate between tumors in locations of high or low language eloquence. We could not show that more patients with highly language-eloquent tumors underwent surgery based on DES language mapping during awake surgery. We could also not differentiate the clinical and radiological outcome of patients with tumors of high or low language eloquence. Without a classification of language eloquence in this large cohort of 147 patients suffering from anatomically language-eloquent gliomas, the scientific community could not draw conclusions on the feasibility of resecting gliomas – primarily defined as language-eloquent – based on DES language mapping during awake craniotomy or nrTMS language mapping. Therefore, we applied the classification of language eloquence with aim of comparing the localization of gliomas and brain tumors per se with a scientifical background.

Apart from that, as stated in the limitations section, of course, this is the first application of the classification. Despite the large number of 147 patients included in our study, the classification must be evaluated by further groups and might undergo modification. Therefore, we encouraged further centers to evaluate the present classification of language eloquence in order to modify it or to confirm its reliability and applicability, as stated in the Discussion section.

Even more, your co-reviewer (Reviewer #1) considers the classification of language eloquence as the strength of the manuscript (Reviewer #1: “The second hypothesis and the strength of the paper is a proposal for a new classification for language eloquence. The authors have to be congratulated for this initiative. As they correctly state, this classification has to be validated by other research groups.”).

Hence, in order to preserve the scientific statement of the present analysis, we would not remove the classification of language eloquence.

Comment 7:

The main issue of this paper is nvTMS and its reliability in surgery. They should highlight this aspect with more detailed data showing the percentage of positive response of the different function of the language area, their concidence with fMRI and intraoperative mapping  that give to us the level of the guarantee we can offer to the patient.   The authors  have presented their  work in a too descriptive way  which is not acceptable for a publication.  A major  revision with more details   is needed  to reconsider this paper.

Answer 7:

Dear reviewer, we thank you for your comments. As mentioned above, we did not perform fMRI in the present study. Additionally, comparisons between nrTMS language mapping data and the gold standard technique DES during awake craniotomy as well as error rates of nrTMS language mappings in patients and healthy subjects have repeatedly been published by several groups during the last decade. Those are vastly cited in our manuscript.

However, in order to answer your comments we added additional data and figures to the manuscript. We think that the incorporation of your comments has significantly improved the manuscript and the essence has become clearer.

References response to reviewers

  1. Corina, D.P.; Loudermilk, B.C.; Detwiler, L.; Martin, R.F.; Brinkley, J.F.; Ojemann, G. Analysis of naming errors during cortical stimulation mapping: implications for models of language representation. Brain Lang 2010, 115, 101-112, doi:10.1016/j.bandl.2010.04.001.
  2. Krieg, S.; Lioumis, P.; Mäkelä, J.; Wilenus, J.; Karhu, J.; Hannula, H.; Savolainen, P.; Weiss Lucas, C.; Seidel, K.; Laakso, A., et al. Protocol for Motor and Language Mapping by Navigated TMS in Patients and Healthy Volunteers; workshop report. Acta Neurochir (Wien) 2017.
  3. Duffau, H. The usefulness of the asleep-awake-asleep glioma surgery. Acta Neurochir (Wien) 2014, 156, 1493-1494, doi:10.1007/s00701-014-2124-7.
  4. Hervey-Jumper, S.L.; Li, J.; Lau, D.; Molinaro, A.M.; Perry, D.W.; Meng, L.; Berger, M.S. Awake craniotomy to maximize glioma resection: methods and technical nuances over a 27-year period. J Neurosurg 2015, 123, 325-339, doi:10.3171/2014.10.jns141520.
  5. Ille, S.; Sollmann, N.; Hauck, T.; Maurer, S.; Tanigawa, N.; Obermueller, T.; Negwer, C.; Droese, D.; Zimmer, C.; Meyer, B., et al. Combined noninvasive language mapping by navigated transcranial magnetic stimulation and functional MRI and its comparison with direct cortical stimulation. J Neurosurg 2015, 10.3171/2014.9.JNS14929, 1-14, doi:10.3171/2014.9.JNS14929.
  6. Giussani, C.; Roux, F.E.; Ojemann, J.; Sganzerla, E.P.; Pirillo, D.; Papagno, C. Is preoperative functional magnetic resonance imaging reliable for language areas mapping in brain tumor surgery? Review of language functional magnetic resonance imaging and direct cortical stimulation correlation studies. Neurosurgery2010, 66, 113-120, doi:10.1227/01.NEU.0000360392.15450.C9.
  7. Picht, T.; Krieg, S.M.; Sollmann, N.; Rosler, J.; Niraula, B.; Neuvonen, T.; Savolainen, P.; Lioumis, P.; Makela, J.P.; Deletis, V., et al. A comparison of language mapping by preoperative navigated transcranial magnetic stimulation and direct cortical stimulation during awake surgery. Neurosurgery 2013, 72, 808-819, doi:10.1227/NEU.0b013e3182889e01.
  8. Tarapore, P.E.; Findlay, A.M.; Honma, S.M.; Mizuiri, D.; Houde, J.F.; Berger, M.S.; Nagarajan, S.S. Language mapping with navigated repetitive TMS: Proof of technique and validation. Neuroimage 2013, 82, 260-272, doi:S1053-8119(13)00512-0 [pii]

10.1016/j.neuroimage.2013.05.018.

  1. Krieg, S.M.; Tarapore, P.E.; Picht, T.; Tanigawa, N.; Houde, J.; Sollmann, N.; Meyer, B.; Vajkoczy, P.; Berger, M.S.; Ringel, F., et al. Optimal timing of pulse onset for language mapping with navigated repetitive transcranial magnetic stimulation. Neuroimage 2014, 100, 219-236, doi:10.1016/j.neuroimage.2014.06.016.
  2. Ille, S.; Sollmann, N.; Hauck, T.; Maurer, S.; Tanigawa, N.; Obermueller, T.; Negwer, C.; Droese, D.; Boeckh-Behrens, T.; Meyer, B., et al. Impairment of preoperative language mapping by lesion location: a functional magnetic resonance imaging, navigated transcranial magnetic stimulation, and direct cortical stimulation study. J Neurosurg 2015, 10.3171/2014.10.JNS141582, 1-11, doi:10.3171/2014.10.JNS141582.
  3. Duffau, H.; Moritz-Gasser, S.; Mandonnet, E. A re-examination of neural basis of language processing: proposal of a dynamic hodotopical model from data provided by brain stimulation mapping during picture naming. Brain Lang 2014, 131, 1-10, doi:10.1016/j.bandl.2013.05.011.
  4. Chang, E.F.; Raygor, K.P.; Berger, M.S. Contemporary model of language organization: an overview for neurosurgeons. J Neurosurg 2015, 122, 250-261, doi:10.3171/2014.10.jns132647.
  5. Ius, T.; Angelini, E.; Thiebaut de Schotten, M.; Mandonnet, E.; Duffau, H. Evidence for potentials and limitations of brain plasticity using an atlas of functional resectability of WHO grade II gliomas: towards a "minimal common brain". Neuroimage 2011, 56, 992-1000, doi:10.1016/j.neuroimage.2011.03.022.
  6. De Witt Hamer, P.C.; Hendriks, E.J.; Mandonnet, E.; Barkhof, F.; Zwinderman, A.H.; Duffau, H. Resection probability maps for quality assessment of glioma surgery without brain location bias. PLoS One 2013, 8, e73353, doi:10.1371/journal.pone.0073353.
  7. Sanai, N.; Mirzadeh, Z.; Berger, M.S. Functional outcome after language mapping for glioma resection. N Engl J Med 2008, 358, 18-27, doi:10.1056/NEJMoa067819.

Round 2

Reviewer 2 Report

The main topic of this paper should be the reliability of TMS and its accuracy in regard to language  mapping , the percentage of positive response in different   language tasks ,  the spatial difference between mapping and   TMS response .

We know that the major reason for an awake craniotomy is the monitoring of the language functions. In our experience we  have a good accuracy in motor area but remains quite questionable the reliability  for language. No comments on this aspect.

The pictures are quite confusing, we should see more cases with preoperative   TMS images  and intraoperative overlapping  with mapping . During resection   I suppose that  the  surgeon has based his decision on the base of the cortical mapping  information. No word about a different strategy.

All the  description is very superficial about the most critical questions of this argument  and consequently this article is  not  helpful for the people who would like to adopt this method.

Finally, the review of critical location for language is completely out of context in such a paper  . It represent an article by itself.